# Testing the Influence of the Material Bonding System on the Bond Strength of Large-Format Tiles Installed on Concrete Substrate under Mechanical Loading

**DOI:** 10.3390/ma13143200

**Published:** 2020-07-17

**Authors:** Libor Topolář, Dalibor Kocáb, Jiří Šlanhof, Pavel Schmid, Petr Daněk, Jaroslav Nováček

**Affiliations:** 1Faculty of Civil Engineering, Brno University of Technology, Veveří 95, 602 00 Brno, Czech Republic; Libor.Topolar@vutbr.cz (L.T.); Dalibor.Kocab@vutbr.cz (D.K.); Pavel.Schmid@vutbr.cz (P.S.); Petr.Danek@vutbr.cz (P.D.); 2Profibaustoffe CZ, s.r.o, Vídeňská 140/113c, 619 00 Brno, Czech Republic; jaroslav.novacek@profibaustoffe.cz

**Keywords:** acoustic nondestructive methods, large format tiles, real-life application, mechanical load, failure

## Abstract

The paper describes an experiment focusing on the way the material system influences the bond strength of large-format tiles installed on concrete substrate during mechanical loading under conditions that correspond to real-life application. This involves a controllable mechanical load applied over an area of a test model while observing its condition using non-destructive methods (ultrasonic pulse velocity test, acoustic emission method, strain measurement, and acoustic tracing). The model consisted of a concrete slab onto which were mounted four different systems with large-format tiles with the dimensions of 3 m × 1 m. The combinations differed in the thickness of the tile, the adhesive, and whether or not a fabric membrane was included in the adhesive bed. The experiment showed that the loading caused no damage to the ceramic tile. All the detected failures took place in the adhesive layer or in the concrete slab.

## 1. Introduction

The history of tiling started in the ancient states of the Mediterranean. The first tiles were used in 4000 BC by the Egyptians; from there they slowly spread to Italy, which can be considered a true treasure trove of beautiful tiles. Both public and ecclesiastical buildings of Medieval Italy were the first to receive decorated tiles, usually of small dimensions. Larger tiles began to be produced in the 12th century in Florence, Sienna, Orvieto, and Faenza. Since the beginning of the 12th century, the development of ceramic tiles had sought to make production more efficient. This meant emphasis on planning, manufacturing, as well as logistics. Thanks to this, the first manufacturing companies began to emerge, some of which still exist to this day [1,2,3].

While at the end of the 20th and beginning of the 21st century, the largest tile was roughly 0.5 m × 0.5 m, nowadays there are tiles with the dimensions of up to 1.6 m × 3.2 m. This marks a significant increase in tile size. The arrival of large-format tiles is one of the main trends of today and goes hand in hand with the modernization of manufacturing [4]. In recent years, the technical properties of the tiles have improved as well; properties such as water absorption, freeze–thaw resistance, wear resistance, or anti-slip safety [5,6]. Large-format tiles are therefore almost exclusively manufactured as the so-called rectified tiles. Their large size is useful, especially in the tiling of large areas, thanks to the minimal number of grout lines in between. At the same time, the thickness of some of the tiles is also being reduced. The most modern technology makes it possible to produce tiles of only 3 mm in thickness. These ultra-thin tiles are usually reinforced with a fibreglass mesh and possess the same technical properties as tiles of the standard thickness of 10 mm, while remaining lighter and thus easier to handle [7,8].

A downside of large-format tiles is the fact that they are very sensitive to the quality and levelness of the substrate, the quality of the substrate priming, adhesives, grouts, and tiling methods.

The goal of this experiment was to determine how the material system between the substrate concrete and the large-format tiles affects the bond strength during mechanical loading. The substrate consisted of a steel-reinforced concrete slab of 6 m × 4 m (nominal dimensions), onto which five large-format tiles of 3 m × 1 m were installed. The slab with the tiles was subjected to area loading using a vacuum chamber. Strain gauges were used to continuously measure the relative strain of both the concrete and the tiles. At the same time, acoustic emission was observed, mainly to record the development of mechanical damage in the concrete and the tiles. After the loading, the integrity of the bond was tested by acoustic tracing. The severity of damage to the tiles themselves was tested using the ultrasonic pulse velocity test. The experiment operated with two types of tiles—one with the thickness of 6 mm and the other 3 mm. The former was chosen because it is one of the most widespread types and the latter because it has been known to crack in real-life application. At the same time, two adhesives of different composition were used to attach the tiles onto the substrate. The first is a lightweight, highly deformable cementitious adhesive modified by a powder polymer binder (12 wt.% of the dry mix) and the other is a deformable cementitious adhesive with extended open time without any lightweight filler and modified by a powder polymer binder (3.5 wt.%).

## 2. Test Methods

### 2.1. Ultrasonic Pulse Velocity Test

This is one of the basic non-destructive methods used to test building materials and structures. It involves the repeated sending of ultrasonic pulses into the material and measuring their velocity. This velocity is influenced by the properties of the material as well as its quality; in materials of poorer quality, the ultrasonic pulse travels at a lower velocity. The ultrasonic pulse velocity test (UPV) can be used to determine e.g., the homogeneity of the material, material properties (modulus of elasticity, compressive strength, etc.), changes in these properties over time (for instance due to degradation), or the presence of cracks or air voids in the material [9,10,11]. The main advantages of UPV include its purely non-destructive character, the possibility to repeat the measurement in the same place at different times (different days, months, or years), or the simple and easy application in a laboratory and on-site [12].

There are two basic applications of the UPV test in civil engineering—the transmission and the pulse echo method [13]. When it comes to testing concrete, ceramics, and similar materials, the more common is the transmission method, which uses two transducers—a transmitter and a receiver. The device then measures the time it takes for the pulse to travel from one transducer to the other. The ultrasonic wave has the highest energy in a direction perpendicular to the position of the transmitter, but in cases where both sides of the tested member are not easily accessible, the transit time of the ultrasonic pulses can be measured in different directions as well. Depending on the position of the transducer, there can be several testing techniques:Direct—the transducers are facing each other directly; this is the best arrangement,semi-direct—the transducers are not placed on opposite sides of the member, or they are not facing each other directly,indirect—the receiver is parallel to the transmitter on the same side of the member, and the measurement is repeated several times with the receiver being shifted by a set distance every time [14].

Indirect testing, which was used in the experiment described herein, is highly sensitive to external interference. This is why it is used when only one side of the structure is accessible (as was the case of this experiment), or when it has been well established that the measured surface is perfectly level and defect-free [15] (as is also true here).

### 2.2. Acoustic Emission Method

Acoustic emission (AE) belongs among the most modern non-destructive methods used in material engineering and fatigue testing. The source of AE may originate from many phenomena depending on the type of material. Most sources of acoustic emission are damage-related [16]. The detection and observation of AE is commonly used for predicting material failure [17]. Its benefit is in the fact that it is a global rather than local method, which means that it observes the overall structure of the material rather than a small area. Moreover, the monitoring can be performed over a short time and is not very labour-intensive. However, a downside is its dependence on the way the material is being loaded [18]. This means that certain discontinuities may not generate a detectable AE at certain types or levels of loading. Unlike most other non-destructive testing methods, AE observes only active defects developing inside the material structure. These defects can only occur when the structure is under load. Passive defects or the structure’s shape have no major influence on localizing AE [19]. The source of AE is the release of energy resulting from stimulation by internal or external stress. An AE event is emitted by irreversible dislocations and degradations created in the material’s micro- and macrostructure. The energy thus released transforms into a mechanical stress pulse propagating through the material as an elastic wave. When the wave reaches the surface, part of it rebounds back and part of it transforms into one or more wave modes. In slabs, waves travel mostly in the form of Rayleigh (or surface) waves. Besides surface waves, there are also e.g., Lamb (or plate) waves. Both of these waves travel at different speeds. The signal that is detected by the AE receiver and converted to electrical current is known as an AE signal [20].

### 2.3. Strain Measurement

Relative strain that develops at a chosen point in the structure or element is measured by a number of devices and sensors which are typically called strain gauges. Strain gauges, the basic types of which are mechanical, wire, and resistance strain gauges, are used mainly for stress analysis and the measurement of important material properties. These measurements always involve relative size change in a given part of the member; i.e. the detection of changes in the measured length (whether an increase or decrease) caused by a change in loading or by the external environment. The length of strain gauges is chosen depending on their design and the particular constraints of the element being measured. This experiment uses resistance strain gauges, which is why the information below applies only to this type.

Measurements of homogeneous materials (such as metals) are typically performed with gauges of 3–10 mm; whereas non-homogeneous materials (e.g., concrete or wood) require longer strain gauges, usually 50–200 mm. How accurately the relative strain is measured (and by extension stress) depends on the quality of the contact of the strain gauge and the material, on the compensation or correction of errors caused by parasitic waves influencing the reading (mainly temperature), and on the necessary calibration and verification of the technique [21,22,23].

### 2.4. Acoustic Tracing

The method studies sub-surface air voids, separated layers beneath the surface, and surface treatment. Its principle involves introducing acoustic energy onto the surface by percussion or by dragging a hard object on the exposed face of the structure. The researcher then observes the acoustic response of the material. The acoustic response can be a ringing, hollow, or crunching sound. This method is used as an auxiliary verification of the bonding strength of the tiling. A ringing response indicates adequate adhesive strength of the bond (a strong enough bonding layer holding the materials together) and satisfactory cohesive strength of the individual materials (the individual cohesion of each separate material). A hollow response is a nearly certain indicator of failures and defects caused by unsatisfactory bond strength (bond failure) [24].

## 3. Experiment

The goal of the experiment was the non-destructive assessment of the quality of the bonding system between large-format tiles and a concrete substrate. The test model consisted of a steel-reinforced concrete slab of 6 m × 4 m × 0.2 m, which held five large-format tiles with the dimensions of 3 m × 1 m attached by different means. Each system was different—they consisted of two different adhesives (from one manufacturer), two different tiles without a reinforcing fibreglass mesh (thickness of 6 mm and 3 mm), and, in one case, the use of a fabric membrane.

### 3.1. Preparation

The Faculty of Civil Engineering of the University of Technology in Brno houses a research centre equipped with a vacuum chamber with the dimensions of 6.18 m × 4.18 m × 0.44 m. It is connected to an Edwards GXS250/2600 dry screw pump (Edwards Group Ltd., Stockholm, Sweden) with a Roots booster mechanism. The pump has enough suction performance to create a vacuum of 700 mbar in the chamber. An advantage of applying a load using a vacuum is that the load is uniformly applied on the entire surface of the specimen. Depending on the performance of the pump, the system can deliver different levels of loading up to 100 kN/m^2^ [25]. The vacuum was created underneath the slab, causing it to deflect downwards and thus introduce strain to bonding systems and tiles.

The floor area of the vacuum chamber was a factor that limited the size of the test specimen. In order to preserve the airtight seal of the chamber (the walls and floor were connected and sealed together), it was not possible to safely remove the formwork of the specimen if it were to be cast inside the chamber. It was thus necessary to design a monolithic slab with lost formwork consisting of 5 reinforced pre-slabs of 1.2 m × 4.05 m × 0.05 m with additional reinforcement and additional concrete layer of 0.15 m on top. The inner surface of the pre-slabs was fitted with 3 strain gauges (Figure 1)—each slab bore 1 gauge (the leftmost and rightmost slabs were not fitted). The additional top layer was made with C 20/25 XC1 concrete with B500B reinforcement, with a concrete cover of 30 mm.

Figure 2 shows the structure of the reinforcement. Until they became fully integral to the slab, the pre-slabs were temporarily supported for additional stability. The supports consisted of two massive frames from which descended suspending rods. The rods were cut once the concrete had hardened. The result was a concrete member that behaved like a monolithic slab. The test chamber has a floor area 60–80 mm smaller than the dimensions of the slab. The gaps were sealed with extruded polystyrene.

After the concrete slab had hardened (i.e. when it reached the age of 28 days), its top surface was ground down by 3–5 mm to create a smooth and level surface (the maximum deviation per 1 m was 1 mm). After that, the surface was cleaned using a powerful industrial-grade vacuum cleaner. Besides these, the surface received no other treatment before the installation of the large-format tiles. The test slab, which had the nominal dimensions of 6 m × 4 m (6 m × 4.05 m to be precise), was divided into five symmetrical test fields. Each field consisted of a bed of adhesive and a large-format tile of 3 m × 1 m with cutouts (see Figure 3). One of the tiles was used in another experiment, which is not part of this paper. This is why this article only discusses measurements performed on four of the tiles designated A through D. The first adhesive, identified as type I, is a highly deformable cement adhesive with an extended open time (class C2ES2 according to EN 12004-1 [26]), with standard setting and a large portion of lightweight filler. This adhesive is heavily modified by powdered polymer binder at an amount of 12 wt.% of the dry mix. The second adhesive, identified as type II, is a deformable cementitious adhesive with extended open time and reduced slip (class C2TES1 according to [26]). It is a normally-setting adhesive without any lightweight filler, modified by a powdered polymer binder at 3.5 wt.% of the dry mix. In the case of tile A, the composite included an additional layer of adhesive and a separating membrane from a synthetic non-woven fabric. Table 1 shows the composition of the material system in each test field. The tiles were installed when the slab reached 35 days of age.

The diagram (Figure 3) shows the precise size and position of the cutouts. The cutouts serve to concentrate stress; i.e. to act as a possible source of failure. Figure 4 shows a diagram of the position of the tiles on the slab as well as the mounting of the slab on the vacuum chamber.

### 3.2. Loading

While mounted on top of the vacuum chamber, the concrete slab was supported by its longer 6-m sides. This meant that when vacuum was introduced to its bottom side, it would deflect at a span of 4 m (3.93 m to be exact), assuming the theoretical shape of a cylinder. The slab was subjected to gradual cyclic loading, where the load was lifted at the end of every cycle. During the first cycle, the load was 10 kN/m^2^ lasting for 10 min, followed by subsequent cycles that were originally planned to be applied in 5 kN/m^2^ increments. However, it was decided prior to commencing the test that the loading cycles would be adjusted depending on how damage and deformations develop. The maximum load was indeed adjusted after the third cycle (both the load and the holding time)—in the end, the structural model was loaded by eight cycles; see Table 2. The loading and unloading rate remained constant throughout the whole test—5 kN/m^2^/min.

The cyclic loading was performed when the adhesive had reached the age of 42 days (the slab was 77 days old by this time). Before the loading, each tile was tested by UPV. Six uniformly spaced measurement lines were drawn on each tile. The lines consisted of 7 points, the first of which marked the position of the transmitter. The remaining 6 indicated the six positions of the receiver during the measurement. This made up a total of 6 measurement points along one line, which thus had the total length of 300 mm. The distance between the point was always 50 mm. Each line was measured three times, which produced a total of 18 pulse velocities per tile. At the same time, the surface of each tile was fitted with two types of sensors—three surface-mounted strain gauges and two AE sensors (AE measuring parameters: frequency range 80–400 kHz; pre-amplifier 35 dB; threshold 400 mV). Two more AE sensors were also placed onto the surface of the concrete slab. The number and spacing of the UPV measurement lines, strain gauges, and AE sensors was identical for all the tiles and is pictured in Figure 3.

## 4. Results and Discussion

### 4.1. Results Obtained during Cyclic Loading

Figure 5a shows the real progress of loading. The graph shows that the original load increment of 5 kN/m^2^ was reduced by half. The reason for this was the formation of the first cracks in the substrate slab. During the following four cycles, the step-increase in strain did not occur (i.e. it corresponded to load), which is why the last loading during cycle No. 8 was increased by the original 5 kN/m^2^ to a total of 35 kN/m^2^. Figure 5b shows a graph depicting the dependence of AE counts on time (blue dots) and the dependence of the relative strain of the slab on time (red curve). The strain gauges were placed beneath the additional concrete layer, which is why the values represent tensile strain. This is an average value made up from data from three strain gauges. However, the record of the relative strain is not complete but ends at a point where one of the gauges stopped measuring; i.e. just before reaching the maximum load during the eighth cycle (the second strain gauge stopped measuring during the holding of the maximum load of the eighth cycle, the third worked until the end of the loading). The highest number of AE overshoots was recorded during the 2nd and 3rd loading cycle, during which the relative strain increased dramatically as well. It therefore appears that the concrete slab suffered the most substantial cracking during these cycles. During the other loading cycles, nothing significant in terms of acoustic emission had occurred.

The values of relative strain in the tiles were measured using three strain gauges glued onto their surface (see Figure 3); the values therefore represent compressive strain. The final values are an average of all three strain gauges. For the sake of clarity, graph axes in Figure 6 show the same range of values. System A, as opposed to the other systems, shows very small relative strain and very low acoustic emissions throughout the loading. A major increase in relative strain occurred only during the last cycle.

Systems B and C show a greater increase in relative strain during the third cycle. It is probable that the slab suffered cracks in these areas and that a part of the tile became detached from the concrete substrate. This is further confirmed by the increased AE counts, which most likely indicate a failure within the adhesive or a failure of the substrate/adhesive or tile/adhesive interface. Another more significant increase in relative strain only occurs during the eighth cycle.

System D showed no response to crack development in the concrete in terms of relative strain or AE counts. The reason may be its lower stiffness (i.e. high flexibility) that comes with its thickness of 3 mm—the tile followed the curvature of the slab’s deflection and, unlike the 6-mm tiles, had a lower tendency to become detached. System D showed a major change during the sixth loading cycle where its AE counts saw a marked increase, and since the seventh cycle it also showed substantial strain.

The analysis of relative strain shows that the stress in system A (containing the membrane) had been markedly reduced since the third cycle. The difference between the system with the membrane and the ones without is visible since the third cycle and very pronounced since the seventh. The membrane reduced the stress transfer from the concrete substrate onto the tile down to a mere 40% of the relative strain of the other tiles (i.e. strain created in tiles that were glued directly onto the substrate). Even when the slab deflected to an extreme 66 mm, tile A suffered no damage. However, during the third cycle, increased AE counts were observed—a clear phenomenon occurred involving the membrane, possibly its elongation or partial damage to its fibres.

Figure 7 and Figure 8 show the dependence of cumulative AE counts on permanent strain per every loading cycle. The loading was divided into three groups of cycles:Cycle 1–3 (solid line)—until the slab had suffered first cracks; the substrate concrete was undergoing the greatest changes and thus the tiles were subjected to the greatest stress;Cycle 3–6 (dashed line)—the loading increment was reduced (a step of 5 kN/m^2^ was changed to 2.5 kN/m^2^); the slab did not suffer significant cracking; smaller changes in relative strain and lesser AE counts were detected;Cycle 6–8 (dotted line)—these cycles brought the substrate concrete close to overall destruction and it was clear that some system suffered a bond failure.

The figures obtained from measuring the slab (Figure 7) clearly show that the first three cycles caused significant damage, accompanied by an extreme increase in cumulative AE counts. There is also a substantial decrease in AE counts, but also in permanent strain during later cycles after the reduction in the loading cycle. The final loading cycles again show an increase not only in AE counts, but also in permanent strain.

All graphs in Figure 8 have the same scale, except for system A where the scale was changed due to the low number of AE counts. System A, which contains the membrane, shows only a small increase in permanent strain and AE counts compared to the other systems. The membrane seems to have a dampening effect on the strain transfer from the substrate. A small inserted graph shows what the results of system A would have looked like if they were plotted to scale. The results in Figure 8 also show a striking similarity between systems B and C. Thanks to its elasticity, system D with the 3-mm tile stresses the adhesive to a smaller degree than systems with the 6 mm tile. The first six loading cycles did not damage the adhesive enough to affect the bond

The curves in Figure 8 were used in a calculation that determined, using linear regression, the slopes for each loading section, which were later compiled into a correlation matrix; see Table 3. It shows a clear correlation between the behaviour of the concrete slab and system D with the 3-mm slab; i.e. the fact, the system copied the slab’s deflections during loading. It also shows an indirect correlation between the slab and systems B and C. This demonstrates the minimal difference between adhesive I and II. The influence of the textile membrane in system A is also visible, as it causes a certain degree of independence on the shape changes in the substrate.

### 4.2. Results Obtained after the Loading

After all the cycles were completed, the surface of all the tiles was examined using acoustic tracing, where a percussive force was delivered by an impact hammer. The surface of the large-format tiles showed no cracks or defects that would be visible by the naked eye. Figure 9 shows a map of damage that was detected by acoustic tracing. System A, with the membrane, showed no damage to the tile or loss of bond strength. The other tiles that were attached directly showed a separation from the substrate around the cutouts. Systems B through D differed in the size of the unbonded areas. Figure 9 shows the percentages of the damaged areas compared to the total area of the tiles.

After the loading, the tiles were again tested using UPV in the same way as before the loading. Figure 10 shows a boxplot of the pulse velocities measured in each tile before and after loading. The statistical analysis of all the systems showed that the ultrasonic pulse velocity before and after the loading did not differ in a statistically significant way, which means that the internal structure of the tiles was not damaged. All the damage thus occurred in the bond or the slab.

## 5. Conclusions

The experiment examined the influence of the bonding system between a concrete substrate and large-format tiles, and observed the bond strength during mechanical loading. The ultrasonic pulse velocity test showed that the loading did not cause damage to the internal structure of the tiles. All the damage that was discovered by other methods thus occurred in the bonding system between the concrete substrate and the tiles, or in the substrate itself.

The results of AE measurements taken during the loading show that the method is useful for observing structural changes in "sandwich" structures. This method was able to indicate the approaching damage in the bonding system in time. The results corresponded with both the outcomes of strain measurement (greater deflection due to cracks in the concrete slab and the increase in permanent strain) and the findings of acoustic tracing, which was performed after the loading was finished.

Brief conclusions for every bonding system:System A, with the separating membrane from unwoven fabric and adhesive I, showed the greatest resistance to the effect of strain in the substrate and an ability of this adhesive bed to protect a large-format tile of 6 mm in thickness even during extreme deflection of the slab. The conclusion is confirmed both by the results of strain measurement performed on the surface of the tile and the lowest recorded AE counts during loading. Acoustic tracing revealed no damage to the tile anywhere throughout its entire area.System B, which consisted of a 6-mm tile glued directly onto the substrate with adhesive I, was the second best option for installing a large-format tile of this thickness.System C, which consisted of a 6-mm tile glued directly onto the substrate with adhesive II, proved to be more susceptible to damage due to substrate deformation than system B, even though fewer AE counts were recorded. During acoustic tracing, this system showed the highest degree of damage from all the systems examined herein, which would explain the lower AE counts (since the detached areas suffer less strain). The type of adhesive used in this case is not suitable for large-format tiles of 3 m length.System D, which consisted of a 3-mm tile glued onto the substrate with adhesive I, showed, to an extent (in this case it was the sixth loading cycle), minimal differences in strain compared to the concrete slab, and AE counts were also lower than in the case of systems B and C. Since the sixth cycle, the stress caused by the deformation of the substrate concrete slab locally exceeded the bond strength of the system and a part of the tile had become detached.

This experiment shows that the choice of adhesive is critical in the application of large-format tiles in high-risk floor structures (i.e. disregarding substrate expansion and contraction, flexible substrate, underfloor heating, shrinking substrate due to insufficiently mature concrete, performing cutouts, etc.), but more importantly, the fabric membrane should be included in the adhesive bed. Furthermore, the condition of large structures should be regularly monitored, mainly to eliminate high financial costs associated with damage. At the same time, it is recommended to use several methods simultaneously to arrive at a clearer idea of what is happening in the material instead of relying on just one method, since its results may be interpreted wrongly.

## Figures and Tables

**Figure 1 materials-13-03200-f001:**
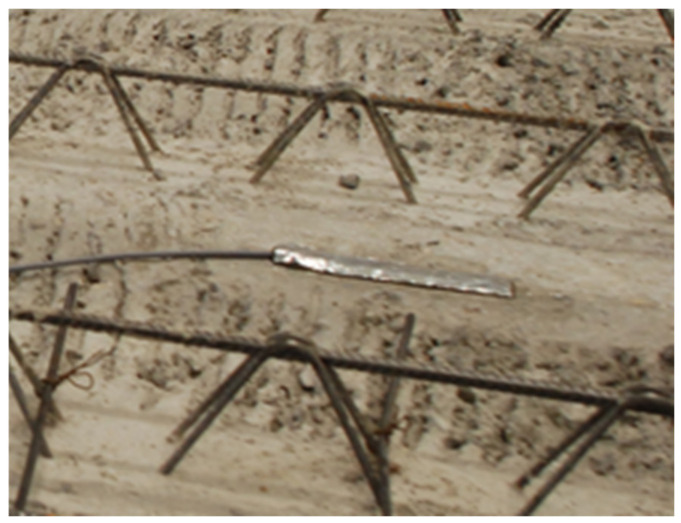
Fitting a strain gauge onto the surface of the lost formwork (reinforced concrete pre-slabs).

**Figure 2 materials-13-03200-f002:**
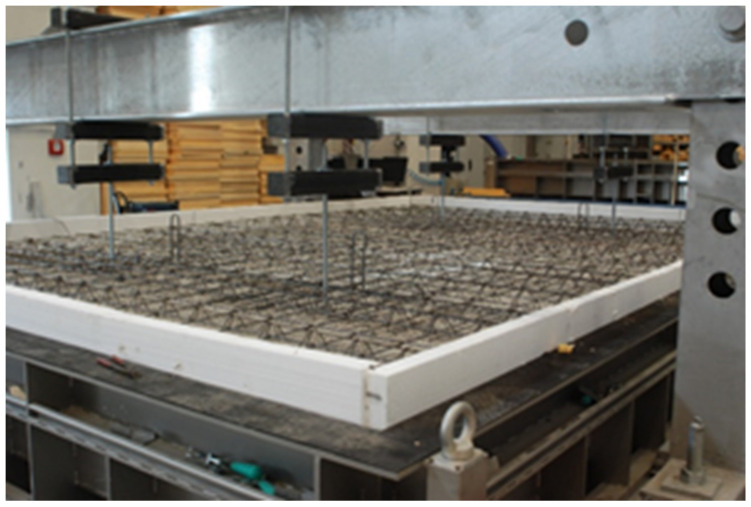
Structure and distribution of the reinforcement.

**Figure 3 materials-13-03200-f003:**
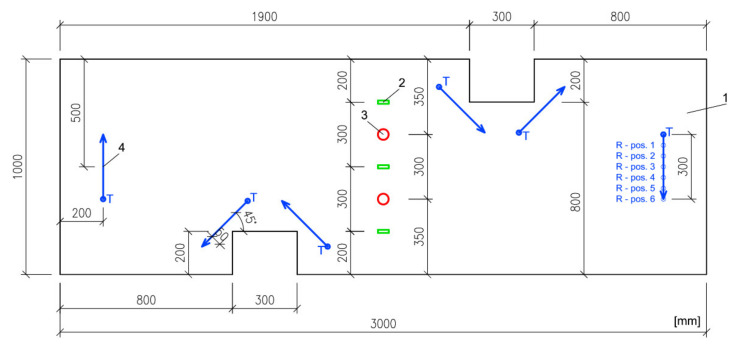
Tile dimensions (1) including the positions of the cutouts, strain gauges (2), acoustic emission (AE) sensors (3) and measurement lines for the ultrasonic pulse velocity (UPV) test (4) with the position of the transmitter (T) and, for illustration, receiver positions indicated at one of the lines (R).

**Figure 4 materials-13-03200-f004:**
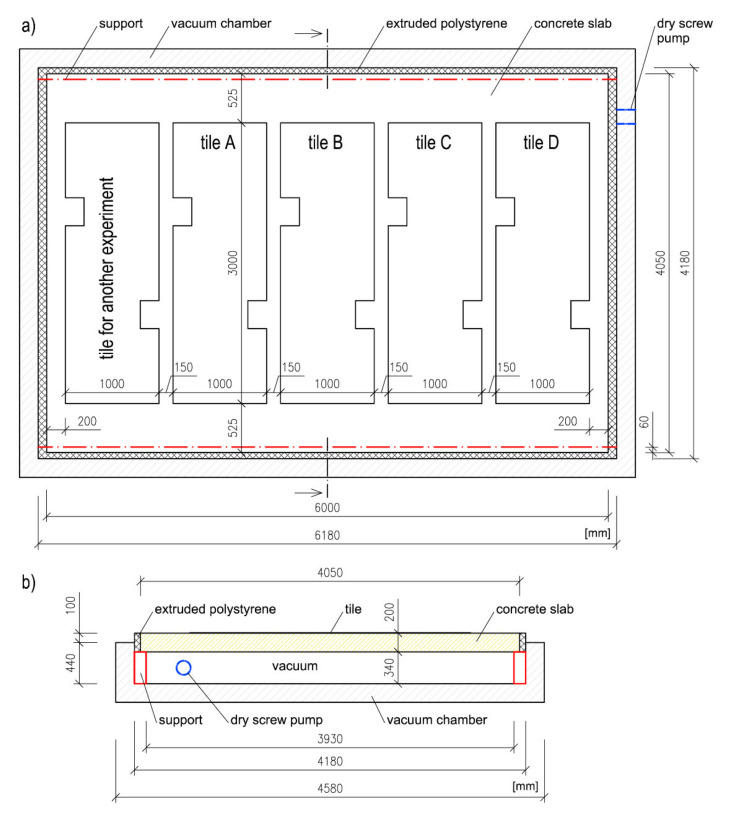
Diagram showing the mounting of the substrate slab with the tiles on the vacuum chamber (**a**) top view, (**b**) cross-section.

**Figure 5 materials-13-03200-f005:**
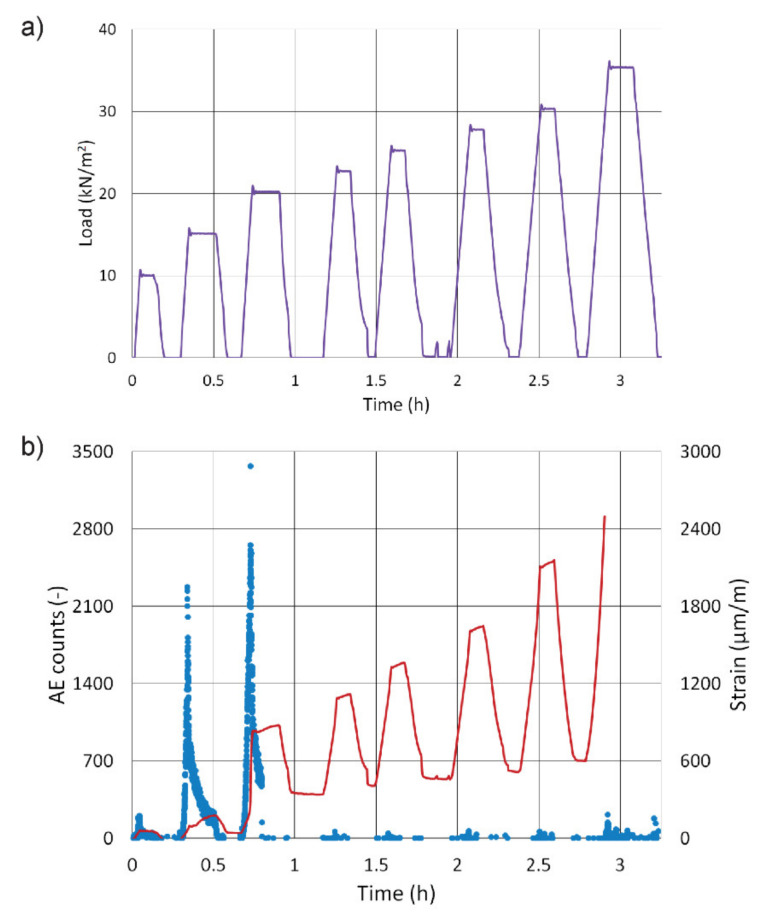
Real progress of loading (**a**); AE counts and relative strain of the concrete slab (**b**).

**Figure 6 materials-13-03200-f006:**
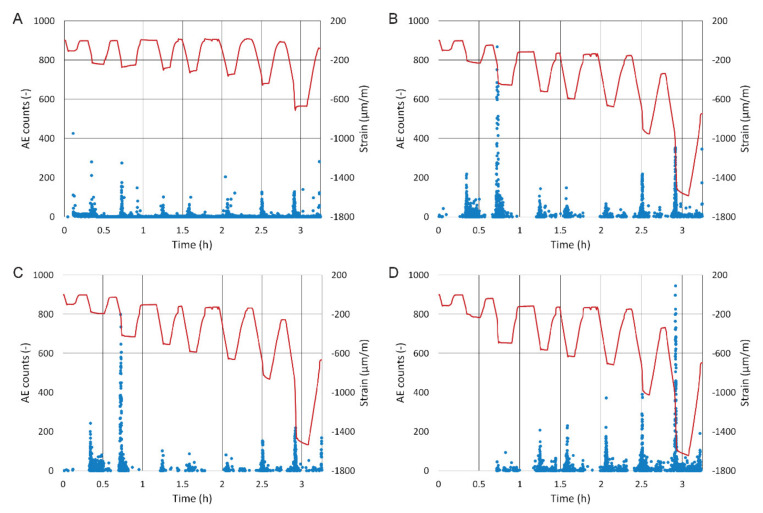
AE counts and development of relative strain in each tile (**A**) material system A, (**B**) material system B, (**C**) material system C and (**D**) material system D.

**Figure 7 materials-13-03200-f007:**
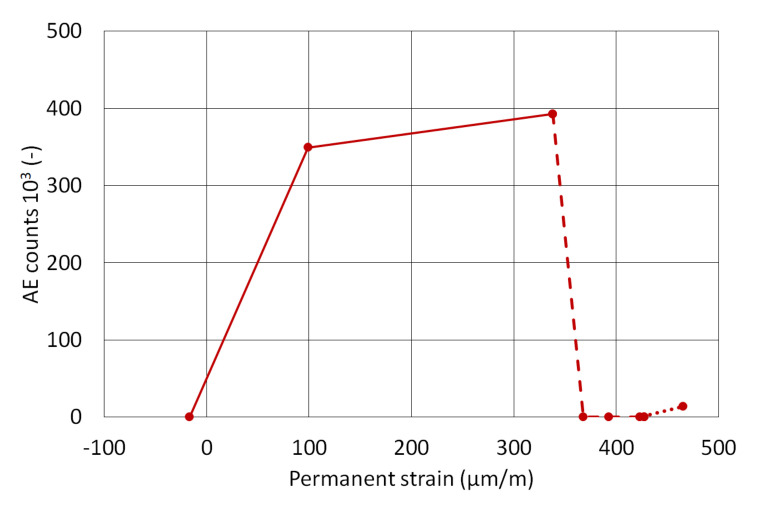
Dependence of AE counts on the development of permanent strain in the concrete slab.

**Figure 8 materials-13-03200-f008:**
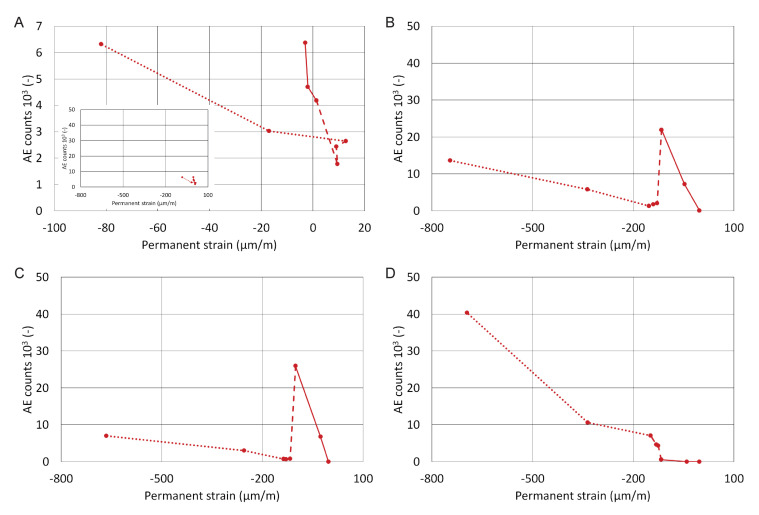
Dependence of AE counts on the permanent strain in (**A**) material system A, (**B**) material system B, (**C**) material system C and (**D**) material system D.

**Figure 9 materials-13-03200-f009:**
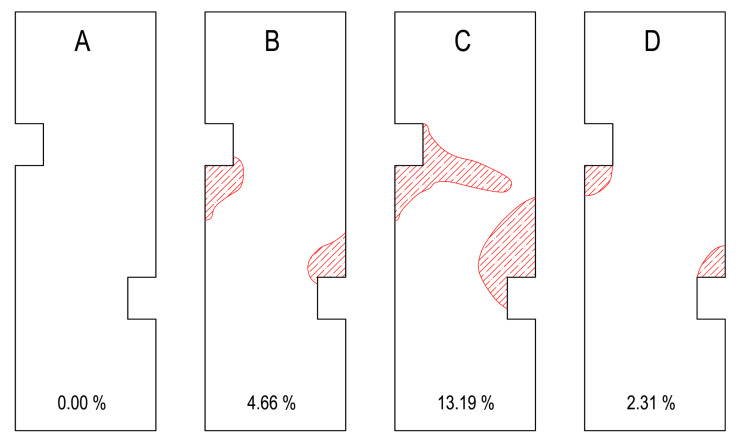
Areas and their percentages where acoustic tracing revealed a loss of bond strength after the eight loading cycles for system A, system B, system C and system D.

**Figure 10 materials-13-03200-f010:**
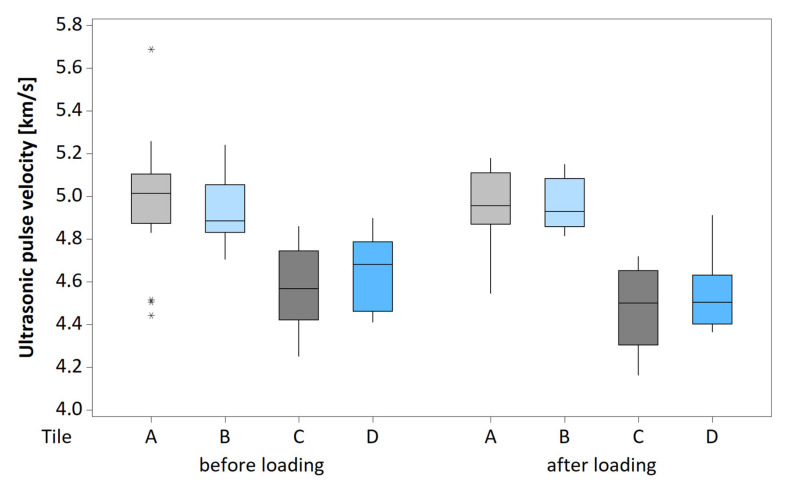
Boxplots of UPV determined by surface measurement on each tile before and after the cyclic loading.

**Table 1 materials-13-03200-t001:** An overview and composition of the test fields.

Field ID	A	B	C	D
**Ceramic element and its thickness**	Iris-Iron Corten 6 mm	Iris-Iron Corten 6 mm	Iris-Iron Corten 6 mm	Levantina Techlam 3 mm
**Adhesive**	Type I	Type I	Type II	Type I
**Fabric membrane**	Yes	No	No	No
**Adhesive**	Type I	-	-	-

**Table 2 materials-13-03200-t002:** List of loading cycles.

Cycle No.	1	2	3	4	5	6	7	8
Maximum load (kN/m^2^)	10.0	15.0	20.0	22.5	25.0	27.5	30.0	35.0
Load-holding time (min)	10	10	10	5	5	5	5	10

**Table 3 materials-13-03200-t003:** A correlation matrix of the groups of loading steps from Figure 8.

	Concrete Slab	A	B	C	D
**Concrete Slab**	1.000	-	-	-	-
**A**	−0.296	1.000	-	-	-
**B**	−0.960	0.550	1.000	-	-
**C**	−0.991	0.418	0.989	1.000	-
**D**	0.985	−0.456	−0.994	−0.999	1.000

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
