# Peer review of "Testing the Influence of the Material Bonding System on the Bond Strength of Large-Format Tiles Installed on Concrete Substrate under Mechanical Loading"

_materials, 2020, doi:10.3390/ma13143200_

Round 1

Reviewer 1 Report

This article deals with the investigation of the influence of the material system between the concrete substrate and the large-format tiles on the bond strength during mechanical loading. A full-scale experiment consisted of a steel-reinforced concrete slab and four large-format tiles was conducted. Each system was different: two types of adhesives, two different tile thicknesses, and in one case the use of a fabric membrane. A cyclic mechanical load in eight increments was imposed by means of a vacuum chamber. Stress and strain states in all four systems were measured using strain gauges and three non-destructive methods: Ultrasonic pulse velocity test, Acoustic emission and Acoustic tracing. The test results are presented and analyzed and the appropriate conclusions are drawn.

The article is interesting, well structured and deserves publication in this journal. In order to improve the quality of the article, the authors should analyze and clarify the following:

Figure 5a shows the cyclic load increments, and Figure 5b shows the measurement results of the relative strain of the concrete slab. For the first six load increments, the expected increase in relative strains is visible. In the seventh and eighth increments, the maximum value of relative strains is equal to that in the sixth increment. It is unusual and completely unexpected. Figure 6 shows the results of measuring relative strains in four large-format tiles and in all cases the expected increase in relative strains in the seventh and eighth load increments. It is also important that the increase in relative strains is very significant in all cases in the last load increment. It is almost impossible for the strains of all four large-format tiles to grow, and the deformations of the concrete slab remain the same in the last two load increments.

The results of Acoustic tracing show that the percentage of the damaged area compared to the total area of tile C is almost three times higher than that of tile B (Figure 9). These two systems differ only in the type of adhesive. The authors therefore conclude that the type of adhesive used in system C is not suitable for large-format tiles. However, the results of Acoustic emission (Figures 6 and 8) show that the AE counts (and permanent strains) of system B are higher than that for system C at the highest loads. This would mean that system C is further from failure than system B. This contradicts the stated conclusion.

Reviewer 2 Report

New technologies for the production and using facing plates of increased size and reduced thickness determine the relevance of the article. Such plates allow finishing work during the construction of buildings with less laboriousness and high productivity, and make their operation more reliable. The article contains a detailed introduction, reflecting both the history of technology for the production of tiles, and methods for their testing. The immediate topic of the article is related to the adhesion strength of such plates installed on a concrete base. The authors describe in detail the conditions and order of the experiments, as well as their results. They analyze technical solutions that improve the reliability of adhesion of plates to the base. The article contains a quantitative statistical assessment of the results of non-destructive testing using ultrasonic sounding. A set of non-destructive testing methods, including ultrasound and tensometric ones, as well as additional acoustic emission measurements, is a positive condition for testing and the reliability of their results. The authors give a number of conclusions that can be used directly in practice. This makes the article especially valuable. However, there are a number of comments related to the insufficiently clear presentation of the materials. An article can be published only after their elimination.

1. The authors in the introduction, as one of the methods for controlling the adhesion of tiles to the substrate, consider acoustic tracing using percussion, but its application is not clearly shown in the further discussion. The following is only about acoustic pulse sounding.

Although the authors try to describe in detail the design of the test setup and the placement of the plate on it, this was unsuccessful. A clear drawing should clarify the verbal description. Without this, many ambiguities arise. For example, how are five slabs 1.2 x 4.05 x 0.05 m placed on a vacuum installation measuring 6.18 × 4.18 × 0.44 m? How is a 6 x 4 m concrete slab tested? It is not clear how a plate with certain dimensions could fit in a vacuum chamber, the dimensions of which were smaller? A clear drawing and a clear description, collected in one place, and not in different places in the text, could eliminate this significant drawback.

In 3.2, the authors write: “This meant that when vacuum was introduced to its bottom side, it would deflect at a span of 4 m, assuming the theoretical shape of a cylinder”. This is not clear, since the principle of operation of the vacuum chamber without a picture is unclear.

4. The schemes of ultrasonic measurements in Fig. 4 are not quite clearly reflected. In addition to the position of the transmitter, the position of the points of location of the receiving transducer should be indicated. The diagram shows the constant locations of the emitter T. To maintain the measurement conditions, it would be better to move the entire circuit at a constant distance between the transmitter and the receiving transducer. It would be desirable to carry out similar measurements in other places of the tiles.

5. The authors did not indicate in which frequency range the acoustic emission was recorded. This is important to know in terms of estimating the size of the area from which the signals were received.

6. The graphs in Fig. 5 show that the number of AE events per second is an informative parameter of acoustic emission. Therefore, the unit of measure on this axis should be impulses / s.

Author Response

  1. The authors in the introduction, as one of the methods for controlling the adhesion of tiles to the substrate, consider acoustic tracing using percussion, but its application is not clearly shown in the further discussion. The following is only about acoustic pulse sounding.

Reply:

The paper discusses two acoustic methods – acoustic tracing (section 2.4) and acoustic emission(section 2.2). The results of both methods are discussed as well – acoustic tracing is in Figure 9, acoustic emission is in Figures 5-8. In the conclusion, the findings from acoustic tracing are used to evaluate each system (lines 393, 401, 403 and 411).

  1. Although the authors try to describe in detail the design of the test setup and the placement of the plate on it, this was unsuccessful. A clear drawing should clarify the verbal description. Without this, many ambiguities arise. For example, how are five slabs 1.2 x 4.05 x 0.05 m placed on a vacuum installation measuring 6.18 × 4.18 × 0.44 m? How is a 6 x 4 m concrete slab tested? It is not clear how a plate with certain dimensions could fit in a vacuum chamber, the dimensions of which were smaller? A clear drawing and a clear description, collected in one place, and not in different places in the text, could eliminate this significant drawback.

Reply:

A diagram has been added to the text showing the positions of the tiles on the slab as well as its mounting on the vacuum chamber (top view and cross-section). It replaces the photograph in Figure 4.

  1. In 3.2, the authors write: “This meant that when vacuum was introduced to its bottom side, it would deflect at a span of 4 m, assuming the theoretical shape of a cylinder”. This is not clear, since the principle of operation of the vacuum chamber without a picture is unclear.

Reply:

The authors believe the new Figure 4 will correct the issue.

  1. The schemes of ultrasonic measurements in Fig. 4 are not quite clearly reflected. In addition to the position of the transmitter, the position of the points of location of the receiving transducer should be indicated. The diagram shows the constant locations of the emitter T. To maintain the measurement conditions, it would be better to move the entire circuit at a constant distance between the transmitter and the receiving transducer. It would be desirable to carry out similar measurements in other places of the tiles.

Reply:

Measurements of UPV were performed by indirect transmission according to EN 12504-4 (cf. Annex A: Determination of pulse velocity – indirect transmission). This method is correct and widely used. If the transmitter and receiver were moved together (to maintain constant distance between them) it would not be possible to measure the UPV. The tested areas of the tiles were chosen to be near the expected points of failure (near the cutouts) and in places where no damage was expected (the edges of the tiles). The authors believe that the number and locations of the testing lines are adequate.

The text has been expanded with information about the length of the line (line 253) and Figure 3 now shows a more detailed diagram of the measurement line.

  1. The authors did not indicate in which frequency range the acoustic emission was recorded. This is important to know in terms of estimating the size of the area from which the signals were received.

Reply:

Information about the parameters of AE measurements was added (line 257).

  1. The graphs in Fig. 5 show that the number of AE events per second is an informative parameter of acoustic emission. Therefore, the unit of measure on this axis should be impulses / s.

Reply:

Fig. 5b shows a graph depicting the dependence of AE counts on time, not AE counts per second. We do not understand this comment. Please specify if possible.

Reviewer 3 Report

The paper reports testing the influence of the material bonding system on the bond strength of large-format titles installed on concrete substrate under mechanical loading. This paper is possibly publishable but should be revised again. For improving the manuscript, it is advisable to address the following comments:

  1. It is better to clearly point out what are non-destructive methods used to observe loading in the abstract.
  2. What is the reason for using 6mm and 3mm thickness of tiles to study? What are the two adhesives? Please explain it in the introduction.
  3. How many times of repeated experiments were measured? There are no error bars of each data points in figure 7 and 8.

Author Response

  1. It is better to clearly point out what are non-destructive methods used to observe loading in the abstract.

Reply:

A list of methods has been added to the abstract (lines 18-20).

  1. What is the reason for using 6mm and 3mm thickness of tiles to study? What are the two adhesives? Please explain it in the introduction.

Reply:

The 6-mm tile (Iris-Iron Corten) was chosen because it is one of the most widespread types of tile, and the 3-mm tile (Levantina Techlam) because it has been known to crack in real-life applications. The commercial availability of the adhesives foes not allow their closer description.

Information about both the tiles and adhesives has been added to the introduction (lines 68-74).

  1. How many times of repeated experiments were measured? There are no error bars of each data points in figure 7 and 8.

Reply:

Given the time scope, technical demands, and high cost (for instance, the vacuum chamber was occupied by this experiment for 11 full weeks), the experiment was performed only once. It is therefore not possible to add error bars to Figure 7 and 8.

Round 2

Reviewer 1 Report

The manuscript has been significantly improved.